# Poly-Gamma-Glutamic Acid Secretion Protects *Bacillus subtilis* from Zinc and Copper Intoxication

Reina Deol,[a] Ashweetha Louis,[a] Harper Lee Glazer,[a] Warren Hosseinion,[b] Anna Bagley,[a] (iD) Pete Chandrangsu[a,b,c]

[a]Keck Science Department, Scripps College, Claremont, California, USA
[b]Keck Science Department, Pitzer College, Claremont, California, USA
[c]Keck Science Department, Claremont McKenna College, Claremont, California, USA

**ABSTRACT** Zinc and copper are essential micronutrients that serve as a cofactors for numerous enzymes. However, when present at elevated concentrations, zinc and copper are highly toxic to bacteria. To combat the effects of zinc and copper excess, bacteria have evolved a wide array of defense mechanisms. Here, we show that the Gram-positive soil bacterium, *Bacillus subtilis*, produces the extracellular polymeric substance, poly-gamma-glutamate ($\gamma$-PGA) as a protective mechanism in response to zinc and copper excess. Furthermore, we provide evidence that zinc and copper dependent $\gamma$-PGA production is independent of the DegS-DegQ two-component regulatory system and likely occurs at a posttranscriptional level through the small protein, PgsE. These data provide new insight into bacterial metal resistance mechanisms and contribute to our understanding of the regulation of bacterial $\gamma$-PGA biosynthesis.

**IMPORTANCE** Zinc and copper are potent antimicrobial compounds. As such, bacteria have evolved a diverse range of tools to prevent metal intoxication. Here, we show that the Gram-positive model organism, *Bacillus subtilis*, produces poly-gamma-glutamic acid ($\gamma$-PGA) as a protective mechanism against zinc and copper intoxication and that zinc and copper dependent $\gamma$-PGA production occurs by a yet undefined mechanism independent of known $\gamma$-PGA regulation pathways.

**KEYWORDS** zinc, copper, poly-gamma-glutamic acid, biofilm, *Bacillus subtilis*

Transition metals ions, such as zinc and copper, are essential for life, yet toxic in excess. Metals have long been appreciated for their antimicrobial properties. Ancient Egyptians used copper salts as early as 2400 BCE as an astringent, food preservative, and disinfectant (1). Since zinc and copper disrupt antibiotic-resistant biofilms, display synergistic activity with other antimicrobials, and kill antibiotic-resistant bacteria, metal-based antimicrobials are currently used in industry, agriculture, and health care as metal-impregnated coatings and surfaces and in combination with traditional antibiotics (2). Furthermore, growing evidence suggest that zinc and copper intoxication play a key role in the host-microbe interaction as antibacterial agents (3).

At elevated concentrations, zinc and copper can outcompete other metals, including manganese and iron, for binding to metal containing proteins (4). Bacteria have evolved diverse mechanisms to prevent zinc and copper toxicity. Under zinc and copper excess, the intracellular level of these ions can be maintained by reduced uptake and/or increased efflux through the regulation of expression and activity of membrane associated metal transporters (5). Zinc and copper can also be detoxified by intracellular sequestration by small proteins or low molecular weight thiols, such as bacillithiol and glutathione (6, 7). The specific proteins and cellular processes poisoned by metals can also be repaired by chaperones or antioxidants or bypassed through the expression of alternative metabolic pathways (8).

During the course of our investigations into the molecular mechanisms underlying bacterial zinc toxicity, we observed that *B. subtilis* colonies became highly mucoid

Address correspondence to Pete Chandrangsu, pchandrangsu@kecksci.claremont.edu.

The authors declare no conflict of interest.

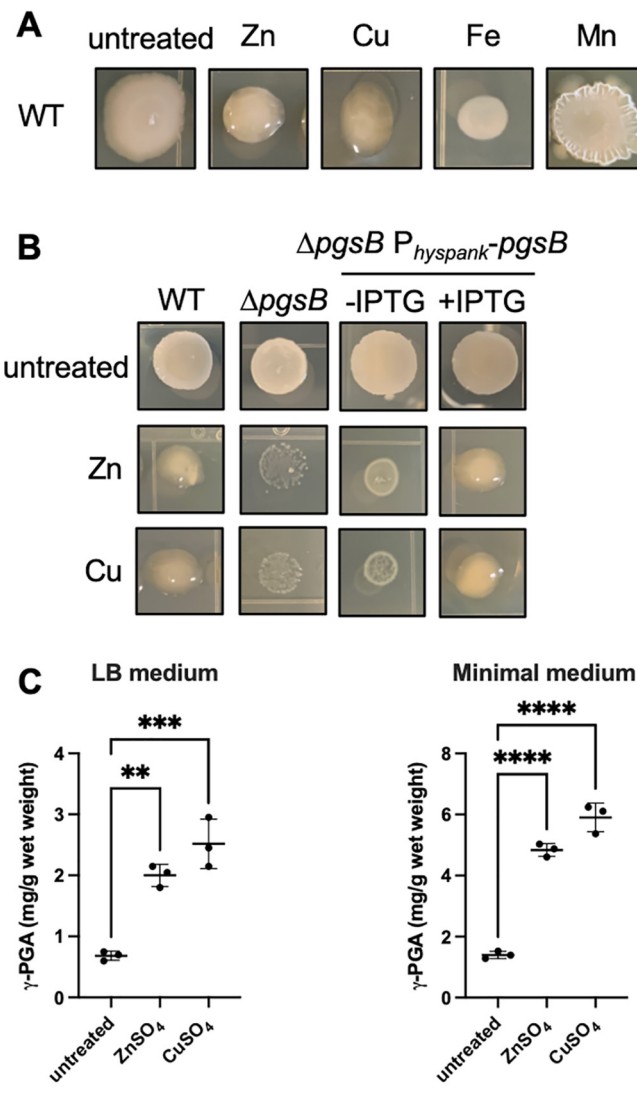

**FIG 1** Zinc and copper induce PGA production. (A) Colony morphology of WT *B. subtilis* 3610 *c*oml in the presence of 250 $\mu$M ZnSO$_4$, 1 mM CuSO$_4$, 2 mM FeSO$_4$, 500 $\mu$M MnCl$_2$. (B) Colony morphology of WT, $\Delta pgsB$, and $\Delta pgsB$ P$_{hyspank}$-*pgs*B grown on LB agar in the presence of 250 $\mu$M ZnSO$_4$ or 100 $\mu$M CuSO$_4$. IPTG (1 mM) was also added to $\Delta pgsB$ P$_{hyspank}$-*pgs*B to induce *pgs*B expression. (C) Concentration of $\gamma$-PGA produced by cells grown in LB and modified minimal medium in the presence of 250 $\mu$M ZnSO$_4$ or 100 $\mu$M CuSO$_4$. The $\gamma$-PGA concentration is expressed as mg of $\gamma$-PGA per gram of cell wet weight. Data shown are the mean and standard deviation from three independent experiments. *P* values were calculated by one-way ANOVA. **, $P \leq 0.01$; ***, $P \leq 0.001$.

when grown in the presence of zinc. Here, we demonstrate that the mucoid phenotype is due to the production of the secreted polymer, poly-gamma-glutamic acid ($\gamma$-PGA), show that $\gamma$-PGA protects *B. subtilis* from zinc and copper intoxication, and provide evidence that the small protein PgsE mediates zinc and copper dependent $\gamma$-PGA production at the posttranscriptional level.

## RESULTS AND DISCUSSION

**Zinc and copper induce *B. subtilis* $\gamma$-PGA production.** While studying the mechanisms of zinc intoxication, we observed that *B. subtilis* colonies displayed a mucoid phenotype (Fig. 1A). *Bacillus subtilis* serves as a model organism for the study of biofilm formation and extracellular polymeric substance (EPS) production (9). There is a growing body of evidence suggesting a link between zinc and copper homeostasis and EPS production. In *B. subtilis*, zinc and copper decrease biofilm hydrophobicity, thereby increasing antibiotic

effectiveness (10). Additionally, excess zinc decreases *Streptococcus pyogenes* hyaluronic acid capsule production by inhibiting phosphoglucomutase, a key enzyme in central carbon metabolism (11). Recent studies on *Escherichia coli* virulence suggest that the presence of capsule may exacerbate bile salt mediated zinc starvation (12).

The EPS is composed of polysaccharides, proteins, nucleic acids, and poly-gamma-glutamic acid (γ-PGA) (13). γ-PGA is a biopolymer consisting of repeating units of L- or D-glutamic acid or a combination of both, and plays a role in virulence, survival during harsh conditions, and sequestration of toxic metal ions (14). Additionally, γ-PGA is a nontoxic, biodegradable compound with utility in medical and commercial applications, microbial biosynthesis of γ-PGA is an extensively studied process (15, 16). Given its anionic charge and affinity for metals, γ-PGA is particular interest as a tool for the bioremediation of toxic metal ions from contaminated water (17). Thus, there is great interest in optimizing growth conditions and understanding the genetic circuitry underlying γ-PGA production.

In *B. subtilis*, the γ-PGA biosynthesis machinery is encoded by the *pgsB*CAE operon. In some bacteria, such as *B. subtilis*, γ-PGA is secreted and released from the cell surface, whereas in others like *B. anthracis* and *S. pneumoniae*, PGA remains attached to the cell as a capsule (18). As a result, when γ-PGA is made and secreted by *B. subtilis*, the colonies produce a mucoid slime layer. Since zinc supplementation is known to increase *B. subtilis* γ-PGA production (19), we hypothesized that the mucoid colony phenotype we observed was due to γ-PGA.

To determine if the mucoidy observed in the presence of zinc is due to increased γ-PGA production, we compared the colony phenotype of wild-type and a *p*gsB deletion mutant in the presence of zinc (Fig. 1B). We observed that in the presence of a sublethal concentration of zinc (250 $\mu$M ZnSO$_4$) wild-type colonies are mucoid and raised, whereas the *p*gsB mutant colonies are dull and flat. Furthermore, the mucoid phenotype is restored to the *p*gsB mutant when *p*gsB is expressed ectopically from an IPTG-inducible promoter (P$_{hyspank}$). These results confirm previous reports linking zinc and γ-PGA production.

To determine if the increase in γ-PGA production is specific to zinc or a general response to metals, we tested for colony mucoidy in the presence of a number of physiologically relevant transition metals (iron, manganese, or copper). Previously reported sublethal metal concentrations (2 mM FeSO$_4$, 500 $\mu$M MnCl$_2$, or 1 mM CuSO$_4$) were individually added to LB agar (20, 21). As previously reported, exposure to manganese leads to a wrinkled colony phenotype (22). Of the metals tested, only zinc and copper are able to induce colony mucoidy, consistant with γ-PGA production (Fig. 1B) As observed with zinc, the colony mucoidy observed in the presence of copper is dependent on the presence of *p*gsB. Additionally, we utilized a spectrophotometric based assay to measure γ-PGA production in liquid media in the presence of zinc or copper. Based on previous studies, we chose 250 $\mu$M ZnSO$_4$ or 100 $\mu$M CuSO$_4$, since these concentrations are sufficient to induce the *B. subtilis* response to zinc and copper intoxication, while having a moderate impact on cell growth (23). When grown in LB or minimal medium (S7$_{50}$ + 5% sucrose), addition of zinc or copper resulted in a 3 to 5-fold increase in γ-PGA secretion (Fig. 1C). From these data, we conclude that zinc and copper increase *B. subtilis* γ-PGA production.

**γ-PGA protects *B. subtilis* from zinc and copper intoxication.** γ-PGA can directly bind to metals, such as zinc, copper, and silver, consistent with a proposed protective role for γ-PGA (24, 25). To determine if γ-PGA production serves as a means of protecting *B. subtilis* from high levels of zinc and copper, we compared the zinc and copper sensitivity of wild-type and a Δ*p*gsB mutant by a disk diffusion assay (Fig. 2A). The Δ*p*gsB mutant is more sensitive to zinc and copper than wild type. Furthermore, the zinc and copper resistance of the Δ*p*gsB mutant is restored when *p*gsB is expressed ectopically from an IPTG-inducible promoter.

We also assessed the contribution of γ-PGA to zinc and copper resistance by monitoring cell growth in LB broth in the presence or absence of zinc or copper (Fig. 2B and

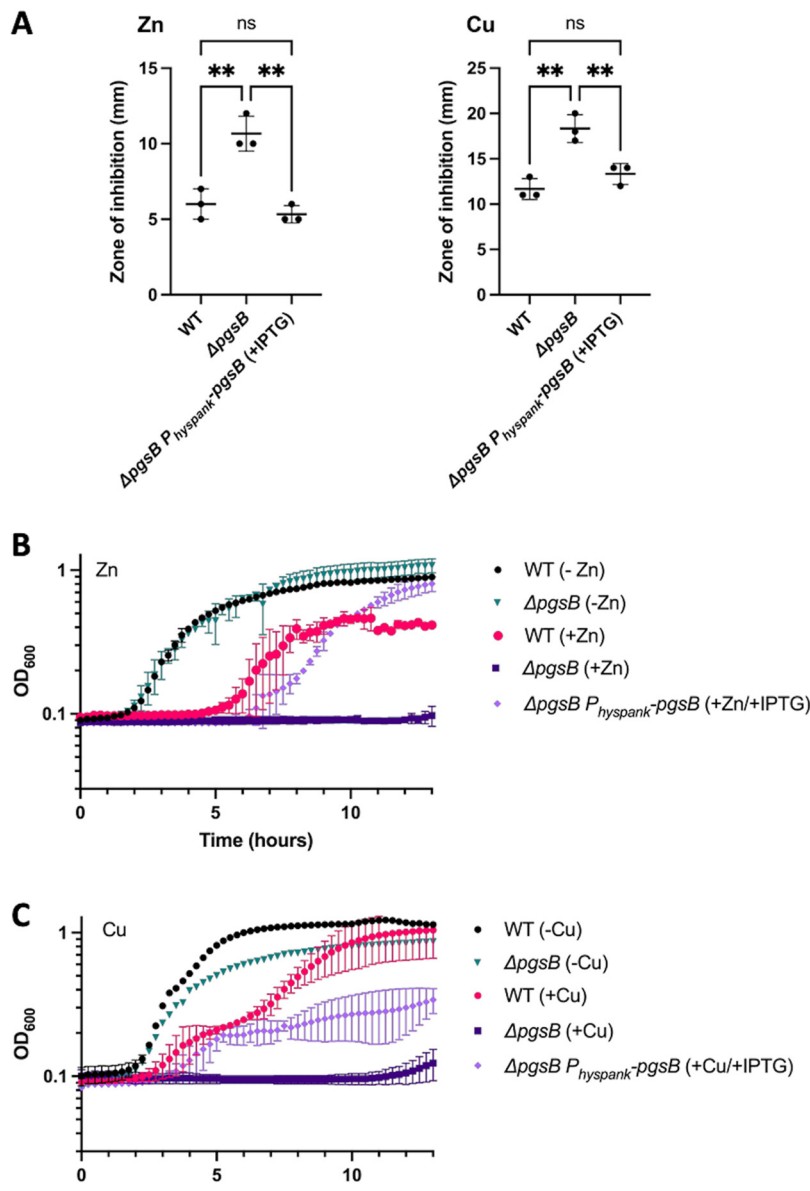

**FIG 2** PGA protects *B. subtilis* from zinc and copper intoxication. (A) Disk diffusion susceptibility test performed on WT, Δ*p*gsB, and Δ*p*gsB P$_{hyspank}$-*p*gsB grown on LB agar. Five $\mu$L of 500 $\mu$M ZnSO$_4$ or 5 $\mu$L of 100 $\mu$M CuSO$_4$ was added to a filter disk. The zone of inhibition was determined by measuring the diameter of the zone of clearing. Data shown are the mean and standard deviation from three different experiments. *P* values were calculated by one-way ANOVA. ns, nonsignificant; **, $P \leq 0.01$. (B and C) Growth curves of WT, Δ*p*gsB, and Δ*p*gsB P$_{hyspank}$-*p*gsB in the presence and absence of either (B) 250 $\mu$M ZnSO$_4$ or (C) 100 $\mu$M CuSO$_4$. IPTG (1 mM) was also added to Δ*p*gsB P$_{hyspank}$-*p*gsB to induce *p*gsB expression. Data shown are the mean and standard deviation from three independent experiments.

C). Wild-type growth is impaired in the presence of 250 $\mu$M ZnSO$_4$ or 100 $\mu$M CuSO$_4$ as indicated by an extended lag phase, decreased growth rate or decreased final OD$_{620}$. In the absence of metal, growth of the *p*gsB mutant is similar to wild type. Upon the addition of zinc or copper, the *p*gsB mutant is unable to grow. When *p*gsB is expressed from an IPTG-inducible promoter, growth of the *p*gsB mutant in the presence of zinc or copper increases. Restoration of growth is to near wild type levels in the case of zinc. In the presence of copper, wild-type like growth is restored during exponential phase. However, the *p*gsB complemented strain is unable to achieve the same final OD$_{600}$ as wild type. Since metals are involved in numerous cellular processes, the

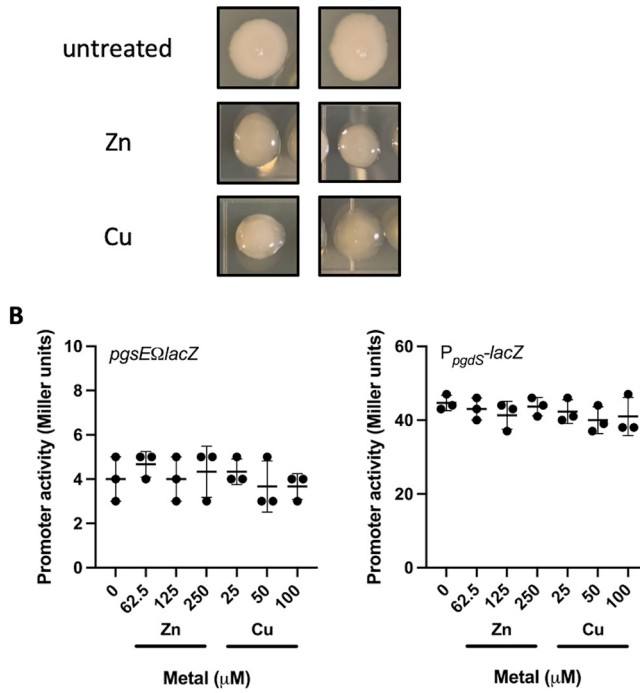

**FIG 3** Zinc and copper dependent PGA production is not controlled at the level of transcription. (A) Colony morphology of WT and ΔdegU grown in LB agar in the presence of 250 $\mu$M ZnSO$_4$ or 1 mM CuSO$_4$. (B) pgs operon (*pgsEΩlacZ*) and P$_{pgdS}$ promoter activity in cells grown in LB media amended with various concentrations of ZnSO$_4$ or CuSO$_4$. Data shown are the mean and standard deviation from three independent experiments. No significant difference was detected between samples by one-way ANOVA.

partial rescue in the presence of copper may be indicative of differences is the physiological processes inhibited by zinc and copper. From the disk diffusion and growth data, we conclude that $\gamma$-PGA protects *B. subtilis* from zinc and copper intoxication on both solid and liquid media.

**Zinc and copper dependent $\gamma$-PGA production is not controlled at the level of transcription.** $\gamma$-PGA production can be regulated by altering the expression of the pgs operon, which encodes the PGA biosynthetic machinery, or of *p*gdS, which encodes a gamma-DL-glutamyl hydrolase involved in $\gamma$-PGA degradation (26, 27). Activation of *p*gsB operon expression involves the DegS-DegU two-component system, the ComP-ComA quorum sensing system, and the SwrA protein. (28). DegS-DegU control the expression of genes involved in many cellular behaviors, including biofilm formation, motility, and competence (29). In response to a variety of stimuli, the cytoplasmic DegS sensor histidine kinase phosphorylates the DegU response regulator (DegU-P), while DegQ enhances DegS-dependent phosphorylation of DegU (28, 30). Once phosphorylated, DegU-P activates pgs operon expression by binding upstream of the *p*gsB promoter (31, 32). ComP-ComA and SwrA indirectly activate *p*gsB operon expression by modulating DegU phosphorylation, in the case of SwrA or by regulating DegQ expression, in the case of ComP-ComA (32–34). Since DegS-DegU is downstream of ComA-ComP and SwrA, we decided to focus on DegS-DegU for further study.

To test if DegU is involved in the zinc or copper dependent production of PGA, we monitored the colony morphology of a *d*egU mutant on LB agar in the presence of zinc or copper (Fig. 3). Similar to wild-type, the *d*egU mutant forms a highly mucoid colony in the presence zinc or copper. We infer that DegU is not required for zinc or copper dependent PGA production.

We next explored the possibility pgs operon or *p*gdS expression may be regulated in response to zinc or copper excess. To determine if expression of the pgs operon or *p*gdS is differentially regulated in response to zinc or copper, we used a $\beta$-galactosidase assay to

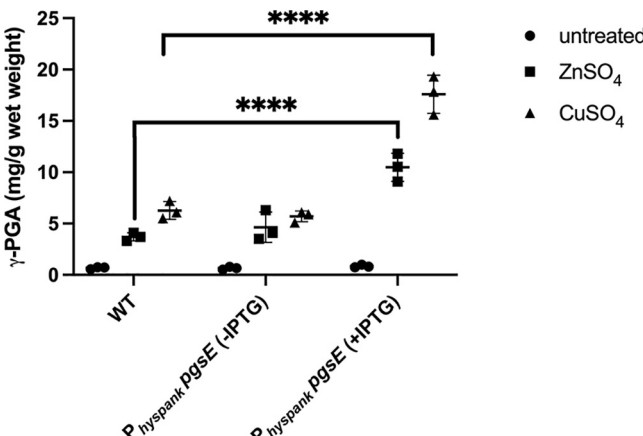

**FIG 4** Overexpression of PgsE enhances Zn and Cu-dependent γ-PGA production. Concentration of γ-PGA produced by WT and PgsE overexpressing cells grown in the presence of 250 $\mu$M ZnSO$_4$ or 100 $\mu$M CuSO$_4$. IPTG (1 mM) was added to induce PgsE overexpression from the *hysp*ank promoter. The γ-PGA concentration is expressed as mg of γ-PGA per gram of cell wet weight. Data shown are the mean and standard deviation from three independent experiments. *P* values were calculated by two-way ANOVA (****, $P \leq 0.0001$).

measure pgs operon and *p*dgS promoter activity in the presence of increasing concentrations of zinc or copper (Fig. 3B). Even when grown in the presence of zinc (250 $\mu$M) and copper (100 $\mu$M) concentrations known to stimulate γ-PGA production, pgs operon and *p*gdS promoter activity did not substantially change. We conclude that zinc and copper dependent γ-PGA production does not occur through changes at the level of pgs or *p*gdS gene expression.

**PgsE (CapE) is required for γ-PGA production in the absence of zinc and copper.** We next explored alternative mechanisms for the zinc and copper dependent regulation of γ-PGA biosynthesis. An intriguing candidate is PgsE, a small, 55 amino acid protein encoded last gene in the pgs operon. The *B. anthracis* PgsE homolog (CapE) is required for γ-PGA synthesis and is believed to play a structural role in the PgsBCAE biosynthetic machinery (35). Interestingly, when PgsE is overexpressed from an inducible promoter on high-copy-number plasmid in the presence of zinc, γ-PGA production by *B. subtilis* (chungkookjang) increases 3-fold compared to the absence of PgsE overexpression (19).

To determine if PgsE also contributes to the copper dependent production we observed, we expressed *p*gsE ectopically from an IPTG-inducible promoter and monitored γ-PGA production in the presence and absence of zinc or copper. When *p*gsE expression is induced in the presence of 250 $\mu$M zinc, γ-PGA levels increase ~2-fold (Fig. 4), similar to previous observations (19). Upon the addition of 100 $\mu$M copper, γ-PGA levels increase ~4-fold compared to the absence of PgsE overexpression. We conclude that PgsE contributes to the zinc and copper dependent regulation of γ-PGA production.

## CONCLUSION

Our studies are consistent with previous work showing that zinc stimulates γ-PGA production by *B. subtilis*. We extend these previous studies and demonstrate for the first time that γ-PGA production is also induced by copper. Based on our results, the mechanism of zinc and copper induction of γ-PGA production is independent of the DegS/U two-component system known to activate pgs operon expression. Furthermore, since neither γ-PGA biosynthesis (*pgsB*CAE) nor degradation (*p*gdS) gene expression is affected by the presence of excess zinc or copper, our data suggest that regulation may be posttranscriptional, likely mediated by the small protein PgsE.

PgsE (*B. anthracis* CapE) is thought to associate with the PgsBCA (*B. anthracis* CapBCA) biosynthetic machinery in *B. anthracis* and when heterologously expressed in *E. coli*. It is tempting to speculate that zinc and copper may be modulating PGA levels

**TABLE 1** Strains used in this study

| Strain | Description | Source |
|---|---|---|
| PC1 | 3610 *com*I (wild-type) | (36) |
| BKE35900 | *trp*C2 Δ*pgsB*::erm | (40) |
| PC2 | PC1 Δ*pgsB* | This work |
| PC3 | PC1 Δ*pgsB amyE*::P$_{hyspank}$-*pgsB* spec | This work |
| BKK35490 | *trp*C2 Δ*degU*::kan | (40) |
| PC7 | PC1 Δ*degU* | This work |
| DS9309 | 3610 *amyE*::P$_{pgdS}$-*lacZ* cat | (41) |
| DS9079 | 3610 *pgsE*Ω*lacZ* cat | (41) |
| PC10 | PC1 *amyE*::P$_{pgdS}$-*lacZ* cat | This work |
| PC11 | PC2 *pgsE*Ω*lacZ* cat | This work |
| PC13 | PC1 *amyE*::P$_{hyspank}$-*pgsE* spec | This work |

by affecting the interaction between PgsE and the PgsBCA protein complex. Our future work will explore the molecular mechanisms by which PgsE increases γ-PGA production in response to zinc and copper. Our work extends our understanding of the protective mechanisms deployed by bacteria in response to metal intoxication and provides insight into the molecular mechanisms underlying γ-PGA production.

## MATERIALS AND METHODS

**Strain and growth conditions.** Strains were grown in Luria-Bertani broth (10 g tryptone, 5 g yeast extract, 5 g/l NaCl) or minimal media (40 mM MOPS ph 7.4, 2 mM potassium phosphate pH 7.0, 2% glucose, 2 g/l (NH$_4$)$_2$SO$_4$, 0.2 g/l MgSO$_4$, 1 g/l sodium citrate, 1 g/l potassium glutamate, 8 mg/l tryptophan, 80 nM MnCl$_2$). When solid agar was used, plates contained 1.5% agar at 37°C. When necessary, antibiotics were added to the growth media at the following concentrations: 1 $\mu$g/mL erythromycin plus 25 $\mu$g/mL lincomycin (mls), 100 $\mu$g/mL spectinomycin, 5 $\mu$g/mL kanamycin, 5 $\mu$g/mL chloramphenicol or 100 $\mu$g/mL ampicillin. One mM of isopropyl $\beta$-d-thiogalactopyranoside (IPTG) was added to the growth medium when appropriate.

**Strain construction.** All strains and primers used in this study are listed in Tables 1 and 2, respectively. To facilitate genetic manipulation, all experiments were performed using the highly competent derivative of *Bacillus subtilis* 3610 carrying a *com*I deletion (3610 *com*I) (36). Chromosomal DNA was used for transformation as previously described (37). To induce natural competence, cells were grown in minimal competence media (1.7 g K$_2$HPO$_4$, 0.52 g KH$_2$PO$_4$, 2 g dextrose, 0.088 g sodium citrate dehydrate, 0.2. g l-glutamic acid monopotassium salt, 1 mL of 1,000× ferric ammonium citrate, and 1 g/l casein hydrolysate and 1% 300 mM MgSO$_4$).

**Colony mucoidy assay.** To evaluate the colony mucoidy phenotype, strains were grown to mid-exponential phase (OD$_{600}$~0.4) in LB broth Five $\mu$L of the culture were then spotted on to freshy poured LB plates containing 1.5% agar that had been dried in the laminar flow hood for 10 min. The spots were allowed to dry an additional 10 min in the laminar flow hood. The plates were then incubated at 37°C for 16–18 h. Images of the colonies were recorded using an iPhone 12 held in a tripod 36 in. from the agar surface.

**PGA quantification.** *B. subtilis* PGA production was measured using a cetryltrimethylammonium bromide (CTAB) assay as previously described (38, 39). Cells were grown in LB broth containing various concentrations of ZnSO$_4$ or CuSO$_4$ in LB media overnight at 37°C for 16–18 h. Cells were then pelleted from 15 mL of culture by centrifugation at 10000 × g for 30 min at 4°C and the supernatant was retained. To precipitate the PGA, 30 mL of ethanol (96%) was added to the supernatant and incubated at 4°C overnight. The precipitated PGA was harvested by centrifugation at 10000 × g for 10 min at 4°C. The resulting pellet was resuspended in 1 mL of distilled water. In a 96-well flat bottom microtiter plate, one hundred $\mu$L of 0.07 M CTAB in 2% (wt/vol) NaOH was added to 100 $\mu$L of the resuspended PGA and mixed gently by pipetting to avoid bubble formation. The OD$_{400}$ of the resulting solution was measured in a Tecan Sunrise microplate reader. A standard curve using purified PGA (Sigma, G1049) was used to calculate the PGA concentration in the broth.

***p*gsB complementation.** The plasmid pDR111 was used to generate the IPTG-inducible *amyE*::P$_{hyspank}$-*p*gsB construct. Briefly, a PCR product containing the *p*gsB coding region was amplified from *B. subtilis* 3610 *com*I chromosomal DNA using primers 1003/1004, digested with SalI and SphI, and cloned

**TABLE 2** Plasmids used in this study

| Plasmid | Description | Source |
|---|---|---|
| pDR244 | *cre ori*$_{TS}$ *amp* spec | (40) |
| pDR111 | *bla a*myE′ P$_{hyperspank}$ spec lacI ′*a*myE | Gift from D. Kearns Lab |
| pPC1 | pDR111 *bla a*myE′ P$_{hyperspank-pk-p}$*gsB* spec lacI ′*a*myE | This work |
| pPC2 | pDR111 *bla a*myE′ P$_{hyperspank-pk-p}$*gsE* spec lacI ′*a*myE | This work |

**TABLE 3** Primer table

| Primer no. | Description | Sequence |
|---|---|---|
| 1001 | pgsB-check-F | TAGGGAAGATTATGTTACATAATGC |
| 1002 | pgsB-check-R | TAAACTGAGTAGTACACCTA |
| 1003 | pgsB-SalI-pDR111-F | ACTGGTCGACATGTGGTTACTCATTATAGC |
| 1004 | pgsB-SphI-pDR111-R | ACTGGCATGCCTAGCTTACGAGCTGCTTTACC |

into the SalI and SphI sites of the pDR111 plasmid. The plasmid was then transformed in to *B. subtilis* 3610 *c*omI. Successful transformation was indicated by spectinomycin resistance and verified by PCR.

**In frame markerless gene deletions.** Gene deletion mutants were constructed from the BKE strains as previously described (40). BKE strains carrying the gene deletion of interest marked by a kanamycin or erythromycin resistance cassette were acquired from the Bacillus Genetic Stock Center (www.bgsc .org). Chromosomal DNA was isolated and transformed in to our wild-type 3610 *c*omI strain. To remove the antibiotic resistance cassette, the temperature sensitive pDR244 plasmid encoding the cre recombinase was introduced into the transformants by selecting for spectinomycin resistance at 30°C. The strains was subsequently cured of the plasmid by growth at 42°C and screened for sensitivity to spectinomycin (loss of the plasmid) and kanamycin or erthyromycin (loss of the antibiotic resistance cassette). Successful gene deletion was verified by PCR using primers 1001/1002 (Table 3).

**$\beta$-galactosidase assay.** A $\beta$-galactosidase assay was used to measure pgs operon and $P_{pdgs}$ promoter activity, as previously described (41). Cells were grown in LB broth in the presence or absence of $ZnSO_4$ (62.5, 125, or 250 $\mu$M) or $CuSO_4$ (25, 50, 100 $\mu$M) at 37°C. One mL of cells were harvested by centrifugation at early stationary phase ($OD_{600}{\sim}1$) and resuspended in an equal volume of Z-buffer (40 mM $NaH_2PO_4$, 60 mM $Na_2HPO_4$, 1 mM $MgSO_4$, 10 mM KCl, and 38 mM 2-mercaptoethanol). Cells were lysed by addition of lysozyme (0.2 mg/mL) and incubation at 30°C for 15 min. Each sample was then diluted in Z-buffer to a final volume of 500 $\mu$L. The reaction was started with the addition of 100 $\mu$L of 4 mg/mL of 2-nitrophenyl $\beta$-d-galactopyranoside (ONPG). Once the sample began to turn yellow, the reaction was stopped with 250 $\mu$L of 1 M $Na_2CO_3$. The $OD_{420}$ was measured, and the specific activity of $\beta$-galactosidase was calculated using the equation (OD420/[reaction time x OD600]) x dilution factor x 1000.

**Growth curves.** To measure the effect of zinc and copper on cell growth, strains were grown to mid-exponential phase ($OD_{600}{\sim}0.4$) and subsequently diluted 1:100 into fresh LB media in the presence or absence of $ZnSO_4$ (250 $\mu$M) and $CuSO_4$ (100 $\mu$M) in a 96-well flat bottom microtiter plate. The plate was then incubated in a Tecan Sunrise microplate reader at 37°C with continuous shaking (high setting) and the $OD_{620}$ was measured every 15 min for 16 h.

**Disk diffusion assay.** Disk diffusion assays were modified from (42). Briefly, a 1:100 dilution of an overnight culture was grown in fresh LB medium at 37°C with 250 rpm shaking until mid-exponential phase ($OD_{600}{\sim}0.4$). A 100 $\mu$L aliquot of cells was then spread on the surface of a LB agar plate (15 mL of 1.5% agar) that had been dried in a laminar flow hood for 10 min. The inoculum was allowed to dry for an additional 10 min in the laminar flow hood. A sterile 6.5 mm Whatman filter disk was then placed on the surface of the agar and $ZnSO_4$ (5 $\mu$L of a 500 mm solution) or $CuSO_4$ (5 $\mu$L of a 100 mm solution) was added to filter disk and allowed to absorb into the disk for 5 min. The plates were incubated at 37°C for 16–18 h, after which the diameter of the zone of inhibition was measured.

## ACKNOWLEDGMENTS

We thank Daniel Kearns (Indiana University) for insightful discussion and generously providing strains.

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
