## [Reviewer comments · Microbiology Spectrum]

Microbiology Spectrum

Poly- γ -glutamic acid secretion protects *Bacillus subtilis* from zinc and copper intoxication

Reina Deol, Ashweetha Louis, Harper Glazer, Warren Hosseinion, Anna Bagley, and Pete Chandrangsu

Corresponding Author(s): Pete Chandrangsu, Scripps College

Review Timeline:

Submission Date:	August 20, 2021
Editorial Decision:	September 20, 2021
Revision Received:	December 22, 2021
Accepted:	February 11, 2022

Editor: Aixin Yan

Reviewer(s): Disclosure of reviewer identity is with reference to reviewer comments included in decision letter(s). The following individuals involved in review of your submission have agreed to reveal their identity: Genwei Zhang (Reviewer #1)

Transaction Report:

DOI: <https://doi.org/10.1128/Spectrum.01329-21>

September 20, 2021

Prof. Pete Chandrangsu
Scripps College
Keck Science Department
925 N. Mills Ave.
KS134
Claremont, CA 91711

Re: Spectrum01329-21 (Poly- γ -glutamic acid secretion protects *Bacillus subtilis* from zinc and copper intoxication)

Dear Prof. Pete Chandrangsu:

Thank you for submitting your manuscript to Microbiology Spectrum. The manuscript has been reviewed by two independent reviewers. Several points need to be addressed before we can consider the acceptance of the manuscript.

When submitting the revised version of your paper, please provide (1) point-by-point responses to the issues raised by the reviewers as file type "Response to Reviewers," not in your cover letter, and (2) a PDF file that indicates the changes from the original submission (by highlighting or underlining the changes) as file type "Marked Up Manuscript - For Review Only". Please use this link to submit your revised manuscript - we strongly recommend that you submit your paper within the next 60 days or reach out to me. Detailed information on submitting your revised paper are below.

Link Not Available

Sincerely,

Aixin Yan

Journals Department
Reviewer comments:

Reviewer #1 (Comments for the Author):

In this manuscript, titled 'Poly-gamma-glutamic acid secretion protects *Bacillus subtilis* from zinc and copper intoxication', Reina Deol and et al. report that *Bacillus subtilis* produces γ -PGA as a protective mechanism in response to zinc and copper excess. Towards molecular mechanism elucidation, the authors provide evidence that the observed zinc and copper dependent γ -PGA production is independent of the DegS-DegQ two component system and claimed that this likely occurs at a post-transcriptional level. Taken together, the paper is well written and easy to follow, and I think this report can broaden the understanding of bacterial metal resistance. Publication can be considered after some revisions are properly addressed.

Main comments:

1. The excitement is diminished after reading the first sentence (line 157) in Discussion (need reference here by the way, ref 34?). If γ -PGA production as a response to zinc excess has been reported before, then the main conclusions of this paper should be tailored, should only focus on copper and then zinc was, in fact, used as a pos control only.
2. To measure γ -PGA production in liquid media, why chose only 250 μ M ZnSO₄ or 100 μ M CuSO₄? I would expect a

concentration titration effect, and a dose-dependent response can provide more information.

3. At the end, molecular mechanism characterization is certainly incomplete, any additional investigation results towards elucidating the mechanism should be added to further strengthen the story.

Reviewer #2 (Comments for the Author):

This is a nice study that nevertheless leaves room for improvement. First question: Why did the authors use LB not minimal salt medium. If the author's aim is to dissect the answer to a single metal minimal salts medium might have been appropriate. How widespread is gamma-PGA in Gram-positive bacteria. This pathway only appears to be present in *Bacillus*. True? The growth curves are nice. This must have been one of the few times where the impact of components of EPS on metal resistance has actually been measured. Is this correct?

The induction experiments are indeed surprising given that there is a clear phenotype. Just out of curiosity, Is this also occurring in minimal salts medium.

Minor points: species in references should be in italic

Staff Comments:

Preparing Revision Guidelines

Please return the manuscript within 60 days; if you cannot complete the modification within this time period, please contact me. If you do not wish to modify the manuscript and prefer to submit it to another journal, please notify me of your decision immediately so that the manuscript may be formally withdrawn from consideration by Microbiology Spectrum.

Here, we will address each reviewers comments individually (**response in bold**).

Reviewer 1:

1. The excitement is diminished after reading the first sentence (line 157) in Discussion (need reference here by the way, ref 34?). If γ -PGA production as a response to zinc excess has been reported before, then the main conclusions of this paper should be tailored, should only focus on copper and then zinc was, in fact, used as a pos control only.

Absolutely! We have modified the text throughout the manuscript to highlight that the zinc effects have been observed previously and that our contribution is to extend this to include copper. We also highlight that our paper is the first to show experimentally that PGA provided protection from zinc and copper intoxication.

2. To measure γ -PGA production in liquid media, why chose only 250 μ M ZnSO₄ or 100 μ M CuSO₄? I would expect a concentration titration effect, and a dose-dependent response can provide more information.

Thank you for your comment. Those values were chosen to be consistent with previous studies on the *B. subtilis* response to zinc and metal intoxication. We have noted this in the text and the appropriate citations have been added (see lines 108-111).

3. At the end, molecular mechanism characterization is certainly incomplete, any additional investigation results towards elucidating the mechanism should be added to further strengthen the story.

We certainly agree that we have not fully characterized the molecular mechanism. In our revision, we include new data to suggest that the small protein PgsE, which was previously implicated in zinc dependent PGA production, is also involved in copper dependent PGA production.

Reviewer 2:

This is a nice study that nevertheless leaves room for improvement.

Thank you for the kind words. We have made every effort to incorporate your suggestions.

First question: Why did the authors use LB not minimal salt medium. If the author's aim is to dissect the answer to a single metal minimal salts medium might have been appropriate.

We have repeated the experiments using minimal salt medium and see a similar metal dependent PGA induction.

How widespread is gamma-PGA in Gram-positive bacteria. This pathway only appears to be present in Bacillus. True?

Yes, this is true. PGA production does seem to be limited to the *Bacilli*. There are some reports that *S. pneumoniae* may also produce PGA, but information is limited.

The growth curves are nice. This must have been one of the few times where the impact of components of EPS on metal resistance has actually been measured. Is this correct?

Yes, this is correct. While other studies have suggested this due to the ability of PGA to bind metals, our study is the first to demonstrate that PGA serves as a bacterial mechanism to prevent metal intoxication.

The induction experiments are indeed surprising given that there is a clear phenotype. Just out of curiosity, Is this also occurring in minimal salts medium.

We have repeated the experiments in a minimal medium and see a similar metal dependent induction of PGA production. Future studies will use this medium as a basis for sorting out the molecular details underlying the phenotype.

Minor points: species in references should be in italic

Thank you for reading the manuscript so carefully!

February 11, 2022

Prof. Pete Chandrangsu
Scripps College
Keck Science Department
925 N. Mills Ave.
KS134
Claremont, CA 91711

Re: Spectrum01329-21R1 (Poly-γ-glutamic acid secretion protects *Bacillus subtilis* from zinc and copper intoxication)

Dear Prof. Pete Chandrangsu:

I am pleased to inform you that your manuscript has been accepted, and I am forwarding it to the ASM Journals Department for publication. You will be notified when your proofs are ready to be viewed.

Sincerely,

Aixin Yan
Editor, Microbiology Spectrum
